# Generic Workflow to Predict Medicine Concentrations in Human Milk Using Physiologically-Based Pharmacokinetic (PBPK) Modelling—A Contribution from the ConcePTION Project

**DOI:** 10.3390/pharmaceutics15051469

**Published:** 2023-05-11

**Authors:** Nina Nauwelaerts, Julia Macente, Neel Deferm, Rodolfo Hernandes Bonan, Miao-Chan Huang, Martje Van Neste, David Bibi, Justine Badee, Frederico S. Martins, Anne Smits, Karel Allegaert, Thomas Bouillon, Pieter Annaert

**Affiliations:** 1Drug Delivery and Disposition, Department of Pharmaceutical and Pharmacological Sciences, KU Leuven, 3000 Leuven, Belgium; nina.nauwelaerts@kuleuven.be (N.N.); julia.macente@kuleuven.be (J.M.); neeldeferm@gmail.com (N.D.); miao-chan.huang@kuleuven.be (M.-C.H.);; 2Simcyp Division, Certara UK Ltd., Sheffield S1 2BJ, UK; 3BioNotus GCV, 2845 Niel, Belgium; rodolfo.h.bonan@bionotus.com (R.H.B.);; 4Clinical Pharmacology and Pharmacotherapy, Department of Pharmaceutical and Pharmacological Sciences, KU Leuven, 3000 Leuven, Belgium; martje.vanneste@kuleuven.be (M.V.N.);; 5Global Research and Development, Teva Pharmaceutical Industries Ltd., Netanya 42504, Israel; david.bibi@teva.co.il; 6Novartis Institutes for BioMedical Research, Novartis, CH-4056 Basel, Switzerland; 7Department of Development and Regeneration, KU Leuven, 3000 Leuven, Belgium; 8L-C&Y, KU Leuven Child & Youth Institute, 3000 Leuven, Belgium; 9Neonatal Intensive Care Unit, University Hospitals Leuven, 3000 Leuven, Belgium; 10Department of Hospital Pharmacy, Erasmus University Medical Center, 3000 CA Rotterdam, The Netherlands

**Keywords:** physiologically-based pharmacokinetic (PBPK) modelling and simulation, in silico, pharmacokinetics, lactation, breastfeeding, human milk, medicines, milk-to-plasma ratio (M/P ratio), daily infant dosage, relative infant dose

## Abstract

Women commonly take medication during lactation. Currently, there is little information about the exposure-related safety of maternal medicines for breastfed infants. The aim was to explore the performance of a generic physiologically-based pharmacokinetic (PBPK) model to predict concentrations in human milk for ten physiochemically diverse medicines. First, PBPK models were developed for “non-lactating” adult individuals in PK-Sim/MoBi v9.1 (Open Systems Pharmacology). The PBPK models predicted the area-under-the-curve (AUC) and maximum concentrations (C_max_) in plasma within a two-fold error. Next, the PBPK models were extended to include lactation physiology. Plasma and human milk concentrations were simulated for a three-months postpartum population, and the corresponding AUC-based milk-to-plasma (M/P) ratios and relative infant doses were calculated. The lactation PBPK models resulted in reasonable predictions for eight medicines, while an overprediction of human milk concentrations and M/P ratios (>2-fold) was observed for two medicines. From a safety perspective, none of the models resulted in underpredictions of observed human milk concentrations. The present effort resulted in a generic workflow to predict medicine concentrations in human milk. This generic PBPK model represents an important step towards an evidence-based safety assessment of maternal medication during lactation, applicable in an early drug development stage.

## 1. Introduction

The World Health Organization recommends exclusive breastfeeding for children up to six months. Breastfeeding plays an important role in the development, health and survival of infants [1]. In addition, breastfeeding is associated with positive effects on the health and well-being of the mother. More than 50% of postpartum women need periodic (e.g., infections, pain) or chronic (e.g., depression, epilepsy) medicines [2]. Although the concentration of most medicines in human milk is expected to be low, there is a lack of robust information [3]. Currently, there is a huge knowledge gap regarding the exposure-related safety of maternal medicines for the breastfed infant. This may expose the infant to (unknown) health risks when medicines are used off-label during lactation. Alternatively, women might discontinue breastfeeding, or delay the initiation of their-much needed pharmacotherapy.

Human lactation studies are costly and associated with ethical and practical challenges. Currently, clinical data available is sparse, and comes mostly from case studies. Non-clinical methods have the potential to generate quantitative data in an early (drug development) stage, and thus inform the label [4]. However, validated non-clinical methods are currently not available. Animal experiments have been performed (mostly in rodents) but have not always been successful in predicting human milk exposure due to species differences in transporters, metabolizing enzymes, and lactation physiology [5]. Similarly, in vitro models have been shown to be promising for the study of medicine transfer across the blood–milk cellular barrier, but characterization of the models has remained very limited.

Physiologically-based pharmacokinetic (PBPK) models are in silico mechanistic models for bottom-up prediction of the pharmacokinetic profile of a medicine. PBPK models require medicine-related input data (e.g., LogP, solubility, permeability), as well as input parameters related to the (patho)physiology of the target population (e.g., blood flow or organ volume). PBPK modelling has already been applied for predicting PK during lactation [6,7,8,9,10].

A first type of model assumes that there is a rapid equilibrium between plasma and human milk. This implies that the concentration of a medicine in human milk can be calculated based on the plasma concentration and milk-to-plasma ratio (M/P ratio). A commonly used method to calculate the M/P ratio is the phase distribution model, assuming that only the unbound and unionized molecular species will cross the blood–milk barrier [7,8,11,12]. An observed M/P ratio can be used as input for the PBPK model if clinical data in plasma and human milk from lactating women are available, but currently these data are not available for most medicines.

Alternatively, the local permeability of medicines at the blood–milk barrier can be implemented in the model. This approach was followed to simulate plasma and human milk concentrations of ondansetron. It was assumed that the permeability of ondansetron to and from human milk was equal, and that the partitioning was assumed to be instantaneous [10]. For some medicines, transporters or carrier proteins present in the membranes of the mammary epithelial cells of the blood–milk barrier might affect the local permeability. This approach was, for example, used to predict medicine concentrations of breast cancer resistance protein (BCRP, *ABCG2*) substrates in human milk. Here, some lactation parameters (e.g., scaling factor for difference between in vivo effective permeability between maternal blood and breast cell and in vitro Caco-2 or MDCK permeability, intrinsic clearance of BCRP for whole breasts) were estimated via parameter optimization [9]. This approach is also called the “middle-out” approach. However, a “middle-out” approach does require the availability of observed clinical data.

The findings reported in this study result from recent efforts within the Innovative Medicines Initiative (IMI) project ConcePTION, which aims to reduce uncertainty about the effects of medicines used during pregnancy and lactation. The hypothesis was that PBPK models can be applied for a full “bottom-up” prediction (i.e., in the absence of observed M/P ratios or measured in vivo concentrations in human milk) of the pharmacokinetic profile of medicines in human milk, and to calculate the daily infant dosage ingested by the infant via breastfeeding. PBPK models were developed for non-lactating adult individuals, and extended to include the lactation physiology. To our knowledge, this is the first study where a generic PBPK workflow for full bottom-up prediction of human milk concentrations in a three-months postpartum population was applied to ten physiochemically diverse medicines.

## 2. Materials and Methods

### 2.1. Model Medicines

Ten medicines were selected based on physicochemical diversity (based on the chemical structures and physicochemical properties as described in Table 1) and the availability of observed in vivo human data in the lactating population (either the availability of data in the literature, or clinical studies that are currently ongoing within the ConcePTION project) for the evaluation of the predictive performance of the PBPK models. The selected medicines were amoxicillin, caffeine, cetirizine, levetiracetam, metformin, nevirapine, sertraline, valproic acid, tenofovir, zidovudine. The Tanimoto coefficients were calculated via ChemMine tools (https://chemminetools.ucr.edu/downloads/ (accessed on 30 January 2023)) based on atom pair and maximum common substructure similarities [13].

### 2.2. Generic Workflow

The generic workflow established for building the lactation PBPK models is described in Figure 1.

The PBPK models were developed using PK-Sim and MoBi version 9.1, available as freeware under the GNU General Public License version 2 (GPLv2) license through Open Systems Pharmacology. An excellent tutorial about the underlying principles and generic procedures for building and evaluating a PBPK models has been published [14]. First, PBPK models for “non-lactating” adult individuals were constructed, and the predictive performance was evaluated using observed data for adult volunteers or patients from the literature. Data extraction from the published literature was done using WebPlotDigitizer (https://automeris.io/WebPlotDigitizer/ (accessed between August 2020 and December 2022), versions 4.3, 4.4, 4.5 and 4.6). Next, the PBPK models were extended to lactation PBPK models. The plasma and human milk concentrations were simulated and compared to observed data for postpartum women. When the time of the sample with respect to the last dose was not mentioned, samples were assumed to be trough samples at steady state (shown as open circles on the plots). The simulated median human milk concentration-time profiles were used to calculate the infant doses via breastfeeding. Plots were created using R for Windows (v. 4.2.2) and Rstudio (v. 2022.07.2+576) (R Foundation for Statistical Computing).

### 2.3. PBPK Models for “Non-Lactating” Adults

The first step was to construct a PBPK model for “non-lactating” adult individuals. Medicine-specific parameters (e.g., physicochemical properties) were combined with study population-specific physiological parameters to predict the pharmacokinetic profile of a medicine in “non-lactating” adult individuals. A first set of observed plasma concentration-time profiles in healthy volunteers and/or patients, containing 1–7 studies with intravenous (IV) and oral (PO) administration (i.e., a training dataset), was used to build the model. For some medicines (levetiracetam, metformin, tenofovir, valproic acid and zidovudine), the fraction excreted in urine (and the fraction metabolized) were also used for model development. Studies with intravenous administration were first used to capture distribution, metabolism, and excretion processes, followed by oral administration to capture absorption. If necessary, parameters were estimated to fit the observed data, implying a “middle-out” approach. The predictive performance of the model was then evaluated with a second dataset, typically containing data from multiple studies (i.e., verification dataset). The generally applied acceptance criterium was a less than two-fold misprediction in plasma area-under-the-curve (AUC) and maximal concentration (C_max_) of medicine for the PBPK models for “non-lactating” adult individuals, although case-by-case judgements were used for data interpretation and acceptance/rejection of the predictions [15]. For a single dose administration, the prediction error was calculated using the AUC from the time of administration until the time of the last observed datapoint. For a multiple dose administration, the AUC was calculated for a dosing interval at steady state. If the observed data was only available for a specific part of the dosing interval, a partial AUC was calculated using Rstudio (DescTools v. 0.99.47.). Details about the model development and evaluation of the predictive performance can be found in Appendix A.

### 2.4. Lactation PBPK Models

The next step was to extend the PBPK models to reflect the physiology of the ‘lactating state’ in three months postpartum population. PK-Sim and MoBi (Open Systems Pharmacology) are an open-source platform that has been extensively used to build PBPK models for different applications. The major benefit of the Open Systems Pharmacology suite is the flexibility to adapt the PBPK model structure in MoBi. The PBPK model for “non-lactating” adult individuals was exported to MoBi, where the spatial structure was adapted to represent a postpartum woman. The final spatial structure (Figure 2) contains a breasts compartment, and a compartment representing the human milk. The workflow followed to add a breasts compartment was based on the tutorial on how to extend a model to a special population in pregnancy [16]. Briefly, the breasts compartment (including physiological parameters, for example the fractions of the different sub-compartments) is a clone of the heart compartment available in PK-Sim.

Information regarding the extent and rate of passage of medicines across the blood–milk barrier is very limited. Therefore, it was decided to rely on a previously reported semi-mechanistic milk–plasma model, which allows for the calculation of the bidirectional clearance values between plasma and human milk based on physicochemical properties of the medicines [17]. This semi-mechanistic model was implemented in the lactation PBPK model structure to allow bottom-up prediction of human milk concentrations. The human milk was directly connected to the plasma compartment, which was consistent with the structure of the semi-mechanistic model [17]. The volume of the milk compartment was fixed to 0.5 L, which was the volume used in the semi-mechanistical model to derive the equations for the bidirectional clearance values [17]. The amount (N) transferred to human milk, and from human milk, were specified in the passive-transport building block using Equations (1) and (2), respectively. It was assumed that medicine-metabolizing pathways were not present in the human milk, based on the fact that there is no evidence for CYP-mediated metabolism.
(1)$$\frac{{\mathrm{dN}}_{\mathrm{milk}}}{\mathrm{dt}}={\;C}_{\mathrm{plasma}}{\;*\;f}_{u,\;\mathrm{plasma}}{\;*\;\mathrm{CL}}_{\sec}\;$$
(2)$$\frac{{\mathrm{dN}}_{\mathrm{plasma}}}{\mathrm{dt}}={\;C}_{\mathrm{milk}}{\;*\;f}_{u,\;\mathrm{milk},\;\mathrm{total}}{\;*\;\mathrm{CL}}_{\mathrm{re}}$$

Hence, in the lactation PBPK models, the transfer of medicines to human milk and reuptake from human milk were parametrized by the respective secretion (${\mathrm{CL}}_{\sec}$) and reuptake (${\mathrm{CL}}_{\mathrm{re}}$) clearance according to the model of Koshimichi et al. (2011) [17]. ${\mathrm{CL}}_{\sec}$ (Equation (3)) is calculated based on the polar surface area (PSA), molecular weight (MW) and the octanol water partition coefficient (LogP) and octanol:buffer (pH 7.4) distribution coefficient (LogD_7.4_). $CL_{re}$ (Equation (4)) is calculated based on LogP and the number of hydrogen bond donors (HBD).
(3)$$\log\left({\mathrm{CL}}_{\sec}\right)=-3.912-0.015\;*\;\mathrm{PSA}+3.367\;*\;\log\left(\mathrm{MW}\right)-0.164\;*\;\log\left(\frac{P}{D_{7.4}}\right)$$
(4)$$\log\left({\mathrm{CL}}_{\mathrm{re}}\right)=2.793+0.179\;*\;\mathrm{LogP}-0.132\;*\;\mathrm{HBD}$$

The lactation PBPK models also required knowledge of the unbound fractions of the medicines in plasma and milk. The equations described by Atkinson and Begg (1990) were used to calculate the (‘total’) free fraction of the medicines in ‘whole’ human milk ($f_{u,\;\mathrm{milk},\;\mathrm{total}}$, Equation (5)). This equation is based on the unbound fraction in skimmed milk ($fu_{\;skimmed\;milk}$, Equation (6)) or the binding to milk protein, and the partition coefficient between milk lipid and ultrafiltrate ($P_{milk}$, Equation (7)). The $f_{u,\;\mathrm{skimmed}\;\mathrm{milk}}$ is based on the relationship between plasma-protein binding and milk-protein binding. $P_{milk}$ was calculated from the octanol:buffer (pH 7.4) distribution coefficient [12].
(5)$$f_{u,\;\mathrm{milk},\;\mathrm{total}}=\frac{1}{\frac{0.955}{f_{u,\;\mathrm{skimmed}\;\mathrm{milk}}}+0.045{\;*\;P}_{\mathrm{milk}}}$$
(6)$$f_{u,\;\mathrm{skimmed}\;\mathrm{milk}}=\frac{f_{u,\;\mathrm{plasma}}{}^{0.448}}{{\left(6.94\;*\;10^{-4}\right)}^{0.448}+f_{u,\;\mathrm{plasma}}{}^{0.448}}$$
(7)$${\mathrm{LogP}}_{\mathrm{milk}}=-0.88+1.29{\;*\mathrm{LogD}\;}_{7.2}$$

All the equations were incorporated in the spatial structure and passive-transport building blocks in MoBi. Therefore, the same workflow can be applied to other medicines requiring only three physicochemical parameters as input (i.e., LogP, PSA and HBD). A step-by-step manual on how to apply this generic workflow to build a lactation PBPK model in PK-Sim and MoBi, Open Systems Pharmacology, as well as the required spatial structure and passive transports are available from Github (https://github.com/translatPK-KUL/LactationPBPK).

PBPK models were developed and evaluated in a three-months postpartum population. Job et al. (2021) previously published a postpartum population, which can be included in PK-Sim [10]. The population contains the physiology of women from the time of delivery (represented by age = 30 years) until two-years postpartum (represented by age = 32 years). In PK-Sim, a three-months postpartum maternal population was created (n = 1000, age = 30.2–30.3 years) from this postpartum population by Job et al. (2021) [10]. The median breast volume and specific blood-flow rate were 1.0 (0.7–1.5) L and 27.0 (26.7–27.3) mL/min/100 g organ. A geometric standard deviation of 1.16 was added for the milk volume (0.5 L) in the population. The administration schedule was adapted to the relevant study designs, as reported in the literature. The median (5th–95th percentile) concentration in plasma and human milk were simulated. For calculating the M/P ratio, as well as the infant dose, a simulation at steady state with a relatively high common-dosing regimen (~worst case scenario) was selected for each compound. The milk-to-plasma (M/P) ratio (Equation (8)) was calculated based on the predicted median area-under-the-curve (AUC) values for a dosing interval at steady state in plasma and milk.
(8)$$M/\mathrm{Pratio}=\frac{{\mathrm{AUC}}_{\mathrm{human}\;\mathrm{milk}}}{{\mathrm{AUC}}_{\mathrm{plasma}}}$$

Information about the dose or time after last dosing for lactation is often missing. Therefore, the evaluation of the predictive performance was case-by-case and mainly based on visual inspection of the predicted versus observed concentration-time profiles in plasma and human milk, and whether the observed data in human milk were within the 5th–95th percentile of the population prediction, in view of the availability and/or quality of the data.

### 2.5. Infant Dose Calculation

The median simulation at steady state, with a relatively high common dosing regimen (~worst case scenario), was selected to calculate the daily infant dose via breastfeeding. The maximum concentration in human milk (C_max, milk_) at three-months postpartum was simulated at steady state. The average concentration in human milk (C_ave, milk_) was calculated by dividing the AUC for a dosing interval at steady state by the dosing interval. The daily human milk intake in infants at this postpartum time interval was assumed to be 150 mL/kg body weight [18]. The daily infant dosage of maternal medicine received via breastfeeding (DID) was calculated using the average, as well as the maximum concentration in human milk (C_milk_ in Equation (9)). Similarly, the relative infant dose (RID) was calculated using the average, as well as the maximal infant daily dosage in comparison to the daily maternal dosage (Equation (10)). The maternal weight was assumed to be 60.3 kg, representing the average three-months postpartum individual from the population by Job et al. (2021) [10]. In addition, if the medicine is also used in children, the daily infant dosage via breastfeeding was also compared to the recommended dose (daily therapeutic infant dosage) prescribed and administered to infants at the same age for therapeutic use (RID_therapeutic_, Equation (11)).
(9)$$\mathrm{Daily}\;\mathrm{infant}\;\mathrm{dosage}\;(\mathrm{DID},\frac{\frac{\mathrm{mg}}{\mathrm{kg}}}{\mathrm{day}})=C_{\;\mathrm{milk}}\;*\;150\;\mathrm{mL}/\mathrm{kg}/\mathrm{day}\;$$
(10)$$\mathrm{Relative}\;\mathrm{infant}\;\mathrm{dose}\;\left(\mathrm{RID},\%\right)=\frac{\mathrm{Daily}\;\mathrm{infant}\;\mathrm{dosage}}{\mathrm{Daily}\;\mathrm{maternal}\;\mathrm{dosage}}\;*\;100\%\;$$
(11)$$\mathrm{Relative}\;\mathrm{therapeutic}\;\mathrm{infant}\;\mathrm{dose}\;\left({\mathrm{RID}}_{\mathrm{therapeutic}},\%\right)=\frac{\mathrm{Daily}\;\mathrm{infant}\;\mathrm{dosage}}{\mathrm{Daily}\;\mathrm{therapeutic}\;\mathrm{infant}\;\mathrm{dosage}}\;*\;100\%$$

## 3. Results

### 3.1. Model Medicines

Table 1 lists the physicochemical properties of the selected medicines. The Tanimoto coefficients were 0.003–0.28 based on atom-pair similarities, and 0.08–0.42 based on the maximum common substructure.

### 3.2. PBPK Model for “Non-Lactating” Adults

A PBPK model for IV administration of amoxicillin was taken from the literature, and extended for oral administration [6]. The metformin PBPK model for IV and oral administration was taken from the literature [19]. For caffeine, the built-in ‘predefined template’ available in PK-Sim was used. For the other compounds, the PBPK models were developed for non-lactating female or male healthy adult and/or patient individuals. Details about the model development and evaluation of the predictive performance are described in medicine-specific reports available in Appendix A. Figure 3 shows the relative predicted/observed ratios for AUC and Cmax for the PBPK models using either the training or verification datasets.

The PBPK models were able to adequately simulate the plasma concentration-time profiles of the medicines in healthy individuals and/or patients. The geometric mean fold errors for AUC and C_max_ were within a two-fold prediction error for all medicines. IndiObserved C_max_ and AUC were all predicted within a two-fold error for amoxicillin, caffeine, cetirizine, levetiracetam, metformin, nevirapine, tenofovir and zidovudine. The PBPK model of sertraline was able to adequately predict the AUC and C_max_ for 24 out of 27 the simulations. It is described in the literature that there is a high inter-individual variability in the pharmacokinetic profiles for sertraline, which is partly explained by the involvement of several enzymes (CYP2D6, CYP2B6, CYP3A4, CYP2C9, CYP2C19, and CYP2E1) with different genotypes in the metabolism [20]. For valproic acid, the PBPK model was able to describe the plasma concentration in 59 out of 62 of the simulations.

### 3.3. Lactation PBPK Models

The PBPK models were extended to lactation PBPK models for all ten medicines. Details about the model development and all plasma concentration-time profile plots simulated for lactation are described in medicine-specific reports available in Appendix A. The predicted secretion and reuptake clearance values are shown in Table 2 for all selected medicine. Figure 4 shows the predicted reuptake clearance values against predicted secretion clearance values for the original dataset reported by Koshimichi et al. (group A–D), as well as for the medicines selected for development of the PBPK models in the present study (group E) [17]. The CL values for the 10 model medicines vary within the range of CL values calculated by Koshimichi et al. (2011) [17]. The lactation PBPK models were able to predict the pharmacokinetic profile in human milk adequately (without estimation or fitting of parameters for lactation) for eight out of ten medicines i.e., amoxicillin, caffeine, cetirizine, levetiracetam, metformin, sertraline, valproic acid and zidovudine. For nevirapine and tenofovir, although the 5th–95th percentile of the population simulation did include most of the observed datapoints, it is clear by visual inspection that there is an overprediction (>2-fold) of the human milk concentration (i.e., the observed milk concentrations are lower). Overall, some datapoints are slightly above the 5th–95th percentile, likely due to the fact that all samples were assumed to be trough samples when information about the time respective to the last dose was missing. The M/P ratios are shown in Table 3.

#### 3.3.1. Amoxicillin

The amoxicillin PBPK model results in a reasonable prediction of the plasma and milk concentrations of amoxicillin, with most observations within the 5th–95th percentile of the population prediction (Figure 5). A dosing regimen of 1000 mg PO three times daily was applied to calculate milk transfer and infant dose. The predicted M/P for amoxicillin was higher than the reported M/P ratios (0.014–0.043) [21]. However, the reported M/P ratios were three single time point M/P ratios around the time of C_max_ in plasma. An AUC-based M/P ratio was not available in the literature. As the milk peak concentration is typically delayed with respect to the plasma peak concentration, this might explain why the predicted AUC-based M/P ratio is lower than the reported M/P ratios. Indeed, if we use the highest measured concentration in plasma (14.60 µg/mL) and human milk (0.81 µg/mL) to calculate the M/P ratio (0.06), the M/P ratio is only 2.5-fold overpredicted. Using non-compartmental analysis (NCA), and assuming that the elimination slope in human milk is identical to the slope observed in plasma, an AUC-based M/P ratio of 0.04 was calculated (4-fold prediction error).

#### 3.3.2. Caffeine

The caffeine PBPK model results in a reasonable prediction of the plasma and milk concentrations of caffeine, with most observations within the 5th–95th percentile of the population prediction (Figure 6). There is quite some variability in the observed data. Importantly, some of the studies were performed at the home of the participants and relied on the subjects to report the time and amount of each dose. In some participants, caffeine was detectable in human milk before they reported the first dose. Therefore, it cannot be excluded that there were differences in the actual dosing and/or sampling times from the times reported by the participants. A dose regimen of 100 mg PO, thrice daily, was applied to calculate milk transfer and infant dose. The predicted M/P ratio was within the observed range.

#### 3.3.3. Cetirizine

The cetirizine PBPK model results in a reasonable prediction of the plasma and milk concentrations of cetirizine, with most observations within the 5th–95th percentile of the population prediction (Figure 7). A dosing regimen of 10 mg PO daily was applied to calculate milk transfer and infant dose. An observed M/P ratio could not be found in the literature. Therefore, the M/P ratio was calculated using the observed steady-state AUC in human milk (0.50 mg*h/L), and the observed plasma AUC in non-lactating adults receiving the same dosing regimen (2.50 mg*h/L) [25,26]. The resulting M/P ratio was 0.2, which is similar to the predicted M/P ratio (0.12).

#### 3.3.4. Levetiracetam

The levetiracetam PBPK model results in a reasonable prediction of the plasma and milk concentrations of levetiracetam, with most observations within the 5th–95th percentile of the population prediction (Figure 8). A dosing regimen of 1500 mg PO twice daily was applied to calculate milk transfer and infant dose. The predicted M/P ratio is within the observed range.

#### 3.3.5. Metformin

The metformin PBPK model results in a reasonable prediction of the plasma and milk concentrations of metformin (Figure 9). After a single dose administration, the elimination phase from the human milk is overpredicted. This may be explained by organic cation transporter involvement in the secretion of metformin into human milk or due to complex partitioning into blood cells. Importantly, this effect is less pronounced after multiple dosing regimens, which are representative of the clinical practice. A dosing regimen of 500 mg PO twice daily was applied to calculate milk transfer and infant dose. After bidaily administration of metformin, most observations are predicted within the 5th–95th percentile of the population prediction. The M/P ratio of metformin is within the observed range, although at the lower end.

#### 3.3.6. Nevirapine

The nevirapine PBPK model results in an overprediction of the human milk concentrations of nevirapine (Figure 10). A dosing regimen of PO 200 mg twice daily was applied to calculate milk transfer and infant dose. The M/P ratio of nevirapine was overpredicted (2- to 13-fold), which is in line with the overprediction in human milk concentration.

#### 3.3.7. Sertraline

The sertraline PBPK model results in a reasonable prediction of the plasma and milk concentrations of sertraline, with most observations within the 5th–95th percentile of the population prediction (Figure 11). There is a high variability in observed data, similar to what was observed in the PBPK model for non-lactating adult individuals. A dosing regimen of PO 50 mg daily was applied to calculate milk transfer and infant dose. The M/P ratio of sertraline is within the broad range of observed M/P ratios.

#### 3.3.8. Tenofovir

The tenofovir PBPK model results in an overprediction of the human milk concentrations of tenofovir (Figure 12). A dosing regimen of 300 mg PO daily was applied to calculate milk transfer and infant dose. The M/P ratio of tenofovir was overpredicted (4- to 16-fold), which is in line with the overprediction in human milk concentration.

#### 3.3.9. Valproic Acid

The valproic acid PBPK model results in a reasonable prediction of the plasma and milk concentrations of valproic acid, with most observations within the 5th–95th percentile of the population prediction (Figure 13). A dosing regimen of 2100 mg PO daily was applied to calculate milk transfer and infant dose. The M/P ratio was within the observed range.

#### 3.3.10. Zidovudine

The zidovudine PBPK model results in a reasonable prediction of the plasma and milk concentrations of zidovudine, with most observations within the 5th–95th percentile of the population prediction (Figure 14). A dosing regimen of 300 PO mg twice daily was applied to calculate milk transfer and infant dose. The M/P ratio was in agreement with the observed range.

### 3.4. Infant Dose Calculation

The infant dose was calculated based on the simulated concentration-time profiles in human milk at steady state. Table 4 shows the daily infant dose (DID, mg/kg/day) and relative infant dose (RID, %) that the infant receives via breastfeeding during maternal pharmacotherapy at steady state. The DID was also compared to a common dosing regimen given to infants for therapeutic reasons, if available (Table 5). The RID received via breastfeeding was very low (<1%) for amoxicillin, metformin, sertraline and tenofovir and low (<10%) for caffeine, cetirizine, valproic acid, and zidovudine. Levetiracetam had a RID slightly above 10% and nevirapine had a RID around 40%. In addition, the DID for all medicines was well below (<25%) the common dosing regimens given to infants for therapeutic reasons. Importantly, if the concentration in human milk is not accurately predicted (i.e., presently the milk concentrations of nevirapine and tenofovir are overpredicted), the infant exposure should be interpreted with caution.

## 4. Discussion

Ten physiochemically diverse model medicines with different elimination pathways were selected to evaluate the predictive performance of newly developed lactation PBPK models by comparing the model-based predictions of the human milk concentration-time profiles with clinical observations from the literature. The lactation PBPK models were able to predict the pharmacokinetic profile in human milk reasonably (without estimation or fitting of parameters for lactation) for eight of the ten medicines: amoxicillin, cetirizine, caffeine, levetiracetam, metformin, sertraline, valproic acid and zidovudine. For nevirapine and tenofovir, there was an overprediction of the human milk concentration. At the same time, from a safety perspective, it is reassuring that no underpredictions for any of the compounds were obtained. A limitation of the current study was the lack of prospective clinical data in the lactating population, to thoroughly evaluate the predictive performance of the PBPK models.

The dose received via breastfeeding for all medicines that were adequately predicted in human milk were low, especially when compared to common dosing regimens given to infants for therapeutic reasons. In addition, whether the infant will develop any (toxic) effect from the medicine does not only depend on the dose received via breastfeeding, but also on the absorption after oral ingestion, as well as on the distribution, metabolism, excretion, and pharmacodynamic profile in the pediatric population. Connecting the PBPK models to infant PBPK models would allow the daily infant dosage via breastfeeding to be used as dosing regimen for a infant PBPK model. This would allow for the prediction of infant systemic exposure based on absorption, distribution, metabolism, and excretion in the infant, taking ontogeny into account. The availability of already existing infant PBPK models describing PK after ‘direct’ dosing in these populations will be an important advantage, although modifications might be needed to represent the breastfed-infant physiology.

This is the first time that lactation PBPK models have been developed according to the same framework in PK-Sim for a total of ten model medicines. The approaches described below can be used to further improve and refine the predictive performance of these models:

First, the quality of the lactation PBPK models is dependent on the quality of the PBPK models for “non-lactating” adult individuals. The PBPK models were developed and evaluated using clinical data for both female and male individuals. Unfortunately, only a single study was found that studied the differences between female and male patients for three medicines (nevirapine, sertraline and valproic acid), and no studies were found for the other medicines. Therefore, it was not possible to fully evaluate the ability of the PBPK models to capture sex differences in PK. The PBPK models can be improved further when additional mechanistic insights (e.g., in vitro data for metabolism) or high-quality clinical data becomes available.

The predictions will also improve by including a more mechanistic model in terms of postpartum physiology, breastfeeding behavior, or human milk composition changes (e.g., pH and lipids) in functions of the postpartum period. Such a dynamic model is currently also available within the simcyp platform. Presently, a relatively ‘basic’ generic PBPK framework was developed, and the focus was on three-months postpartum. However, the same approach can easily be applied to other postpartum periods, although it is not recommended to extend this model to a very early postpartum stage (e.g., immature blood–milk barrier and differences between colostrum and mature milk). In this context, it has been suggested that infants of 2–3 weeks of age are likely at the highest risk in terms of medicine exposure. Indeed, while milk-volume intake is reaching high levels [68], the maturation of elimination pathways in general displays slower ontogeny [69].

Additional refinement of the present PBPK models will be based on permeability coefficients generated in an in vitro model for the blood–milk barrier. This will improve predictions for medicines with transport routes (e.g., active transport) or physicochemical properties that are not fully captured by the semi-mechanistic model from Koshimichi et al. (2011) [17] (e.g., the model is quite sensitive to the input value for LogP and LogD). These data are currently being generated with a newly developed in vitro model based on human mammary epithelial cells [4]. Furthermore, in vivo data are generated in the Göttingen minipig, which are expected to reveal additional mechanistic insights.

Finally, parameter estimation in the lactation PBPK models is likely to further improve the simulations of the lactation PBPK models. However, this requires availability of (high-quality) clinical data for training and verification, often not (yet) available. Information regarding the dosing regimen, postpartum period and time of sampling is often incomplete in the literature. Moreover, only a single study was found for some medicines (e.g., amoxicillin and cetirizine).

The workflow developed here can be applied to predict the human milk concentrations of any small molecule medicine. A potential criticism of this PBPK-based method is the complexity of the underlying models, especially when clinical data are available. In our view, this criticism is currently unjustified, unless rich datasets provide maternal systemic exposure, average concentrations in milk, and infant systemic exposure. A retrospective analysis of the medicines included in the LactMed database showed that 51% of the medicines had no information about their use during lactation [3]. Only 2% of the medicines had data in all categories, supporting the use during lactation. In addition, the quality of the available data is often low, due to inconsistencies between trials. Indeed, most datasets in the ‘lactating’ population would not even allow identification of simple compartmental models. Instead, borrowing strength (prior knowledge) from physiology and pharmacology allows researchers to make dose-exposure predictions, even without the availability of any clinical data.

## 5. Conclusions

The lactation PBPK models resulted in reasonable predictions of maternal plasma and human milk concentration-time profiles for eight medicines, while overprediction (2- to 16-fold) of the concentration-time profiles in human milk was observed for nevirapine and tenofovir. From a safety perspective, it is important that none of the models resulted in an underprediction of the milk levels for the given maternal dosing regimen. The workflow for the PBPK model development and evaluation can be used to simulate the human milk concentration, and to calculate predicted infant doses for virtually any medicine. The framework established here represents an important step towards an evidence-based safety assessment of maternal medication for breastfed infants. A particular strength of this approach is that unique quantitative information on infant exposure to maternal medicine can be generated in an early (drug-development) stage, i.e., before clinical data becomes available.

## Figures and Tables

**Figure 1 pharmaceutics-15-01469-f001:**
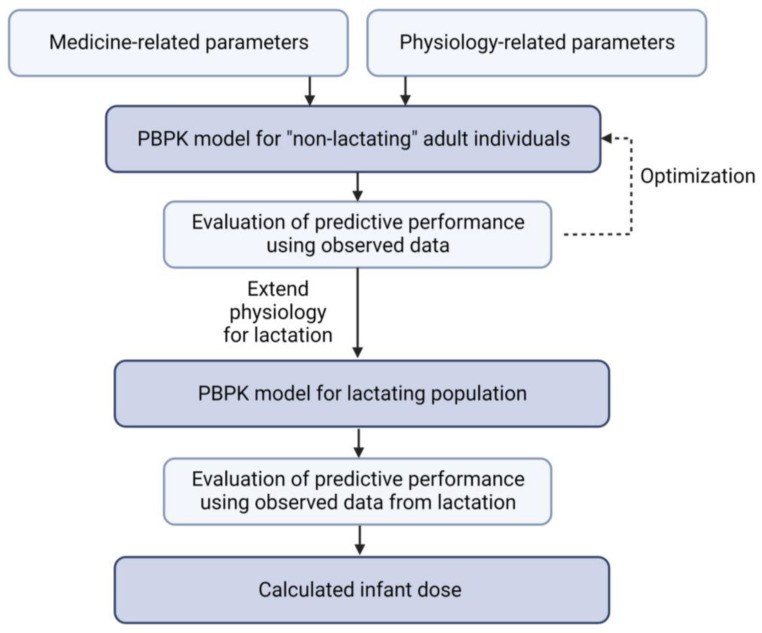
Workflow used in the present study to develop and evaluate the lactation PBPK models.

**Figure 2 pharmaceutics-15-01469-f002:**
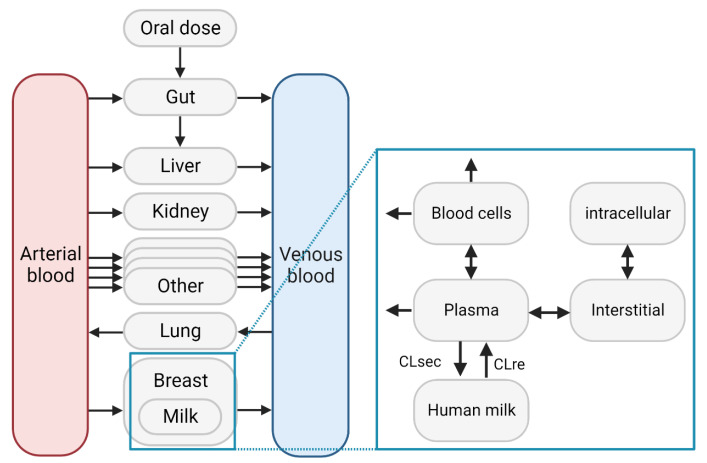
Model structure for the lactation model in PK-Sim and MoBi. The median (interquartile range) breasts volume and specific blood flow rate were 1.0 (0.7–1.5) L and 27.0 (26.7–27.3) mL/min/100 g organ. The blood cell volume was specified as the fraction vascular (0.14) multiplied with the hematocrit and the total breasts volume. The interstitial volume was the fraction interstitial (0.10) multiplied with the total breasts volume. The intracellular volume was the fraction intracellular (0.76) multiplied with the total breasts volume. The plasma volume was the fraction vascular (0.14) multiplied with 1-hematocrit and the total breasts volume. The human milk volume was 0.5 L and a geometric standard deviation of 1.16 was assumed for population simulations.

**Figure 3 pharmaceutics-15-01469-f003:**
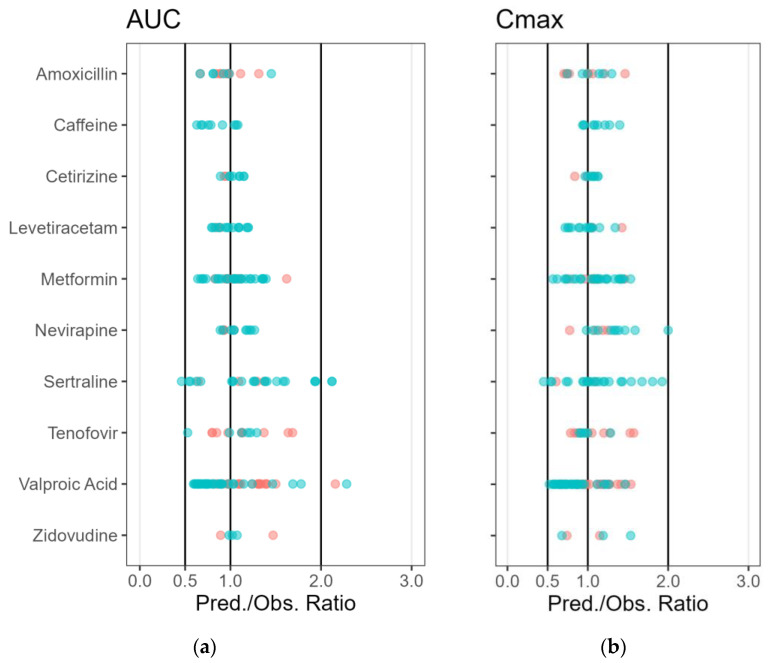
Individual predicted/observed ratios for: (**a**) the area-under-the-curve from the first to the last observed concentration in plasma (AUC); and (**b**) maximal plasma concentration (C_max_) of the selected medicines using developed physiologically-based pharmacokinetic (PBPK) models for “non-lactating” adult individuals. Individual predicted/observed ratios are shown for model building (red circles) and model verification (blue circles) data. Black lines represent the 0.5- and two-fold prediction error ratio. The geometric mean fold error for AUC and C_max_ was within two-fold prediction error for all medicines.

**Figure 4 pharmaceutics-15-01469-f004:**
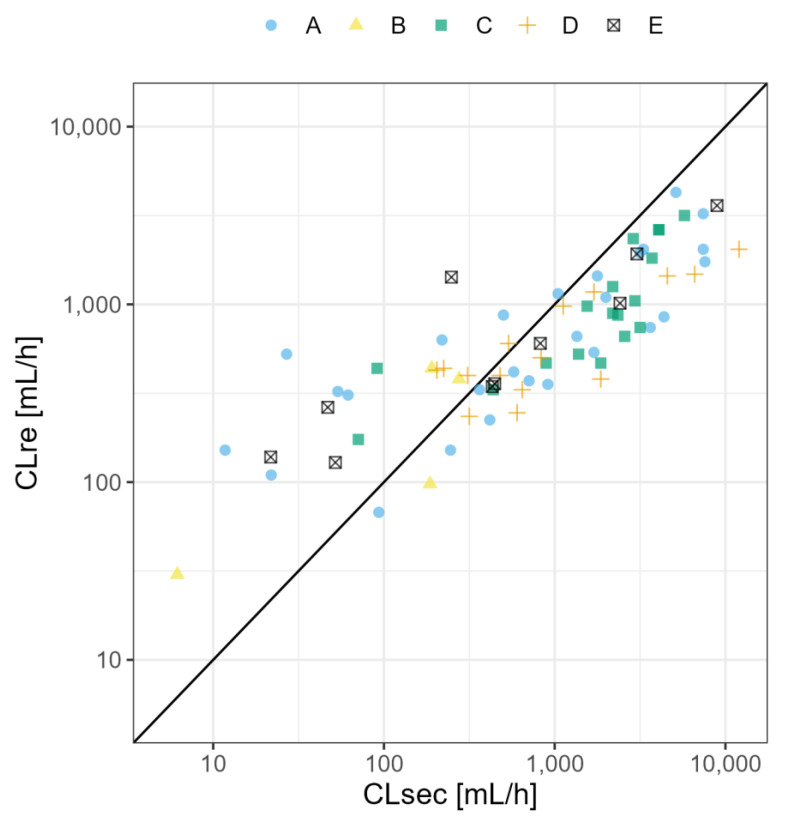
Predicted reuptake clearance (CL_re_) and secretion clearance (CL_sec_) values for medicines included in the original dataset as predicted by Koshimichi et al. (2011) in the original publication [17] for medicines with: (**A**) net reuptake (i.e., total free fraction multiplied with reuptake clearance) values < 5000 mL/h, total free fraction in milk determined experimentally and no evidence for transporter-mediated transfer; (**B**) net reuptake values < 5000 mL/h, total free fraction in milk determined experimentally and transporter-mediated transfer; (**C**) net reuptake values < 5000 mL/h and total free fraction in milk predicted; (**D**) net reuptake > 5000 mL/h; and (**E**) calculated reuptake and secretion clearance values for medicines for which PBPK models were developed in the present study.

**Figure 5 pharmaceutics-15-01469-f005:**
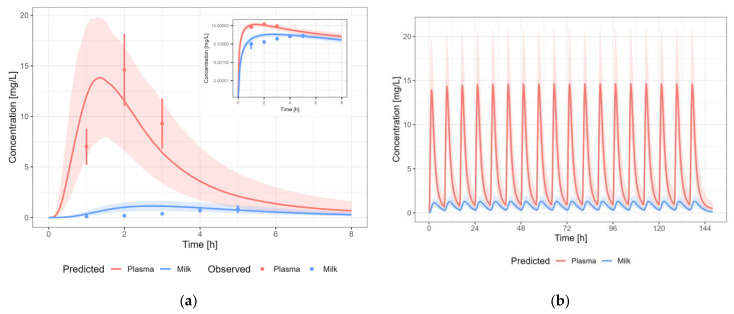
Median (solid line) and 5th–95th% prediction interval (shaded area) for: (**a**) oral administration of amoxicillin 1000 mg as single dose [21]; and (**b**) oral administration of amoxicillin 1000 mg thrice daily in plasma (red) and human milk (blue). Circles represent observed data from the literature with standard deviation (error bars) [21].

**Figure 6 pharmaceutics-15-01469-f006:**
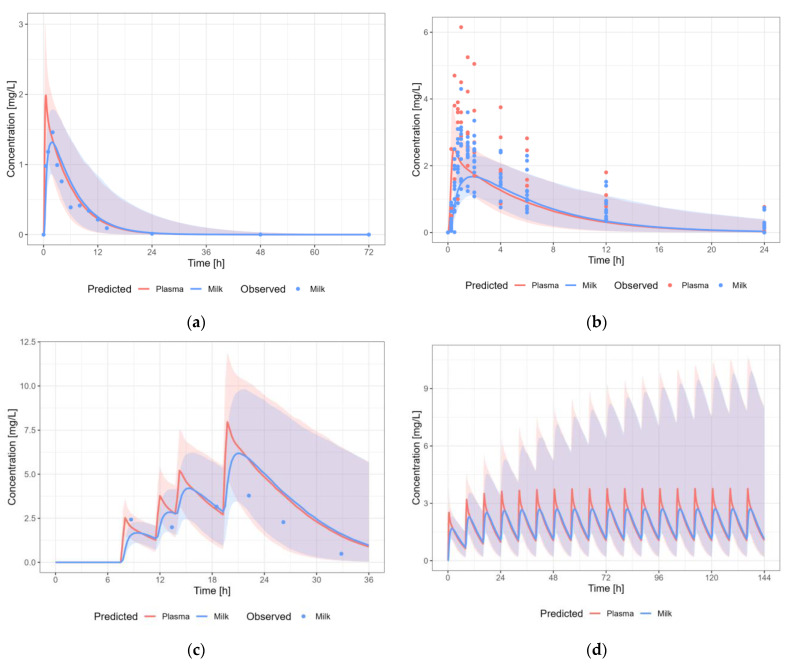
Median (solid line) and 5th–95th% prediction interval (shaded area) for (**a**) oral administration of caffeine 80 mg as single dose [60]; (**b**) oral administration of caffeine 100 mg as single dose [23]; (**c**) oral administration of 100 mg caffeine as multiple dose [61]; and (**d**) oral administration of caffeine 100 mg thrice daily in plasma (red) and human milk (blue). Circles represent observed data from the literature [23,60,61].

**Figure 7 pharmaceutics-15-01469-f007:**
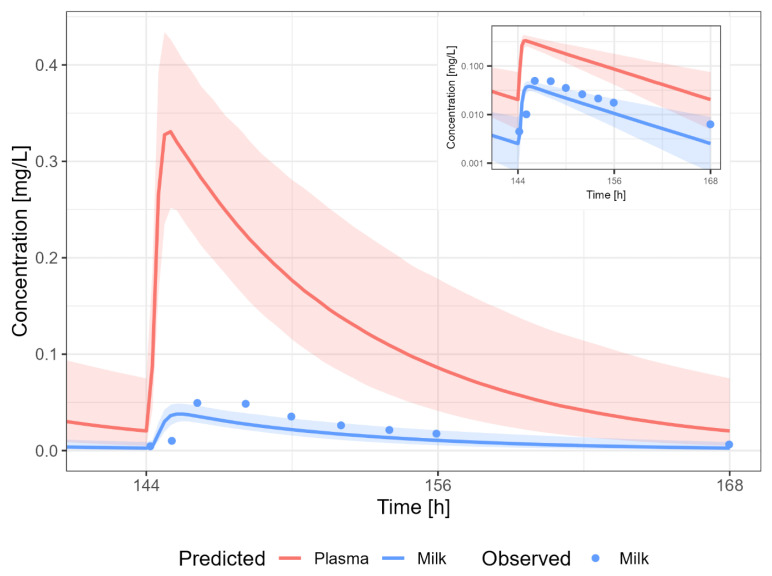
Median (solid line) and 5th–95th% prediction interval (shaded area) for oral administration of cetirizine 10 mg daily in plasma (red) and human milk (blue). Circles represent observed data from the literature [25].

**Figure 8 pharmaceutics-15-01469-f008:**
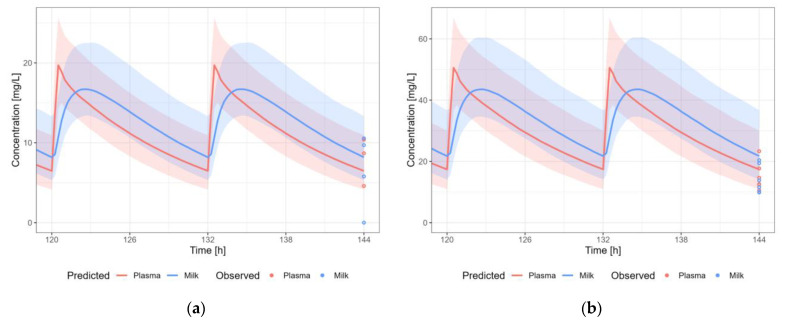
Median (solid line) and 5th–95th % prediction interval (shaded area) for: (**a**) oral administration of levetiracetam 1000 mg as multiple dose [30,62]; (**b**) oral administration of levetiracetam 2500 mg as multiple dose [28,30]; (**c**) oral administration of 2525 mg levetiracetam as multiple dose [27]; and (**d**) oral administration of levetiracetam 1500 mg bidaily [28,30] in plasma (red) and human milk (blue). Circles represent observed data from the literature [27,28,30,62]. Datapoints for which the time of sampling with respect to the last dose was not reported are indicated with open circles.

**Figure 9 pharmaceutics-15-01469-f009:**
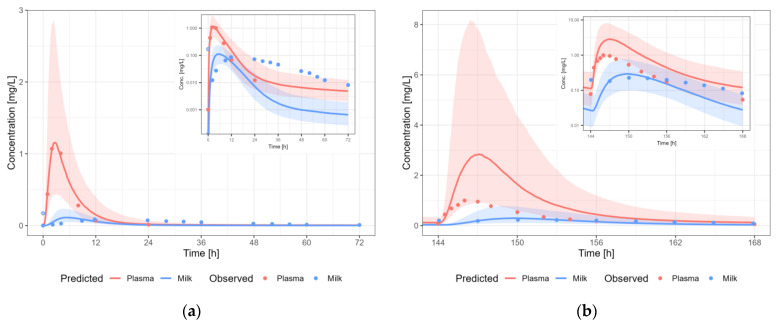
Median (solid line) and 5th–95th % prediction interval (shaded area) for: (**a**) oral administration of metformin 500 mg as single dose [33]; (**b**) oral administration of metformin 1500 mg/day PO [32]; (**c**) oral administration of metformin 500 mg thrice daily [34]; and (**d**) oral administration of metformin 500 mg bidaily [31,33,63] in plasma (red) and human milk (blue). Circles represent observed data from the literature with standard deviation (error bars) [31,32,33,34,63]. Datapoints for which the time of sampling with respect to the last dose was not reported are indicated with open circles.

**Figure 10 pharmaceutics-15-01469-f010:**
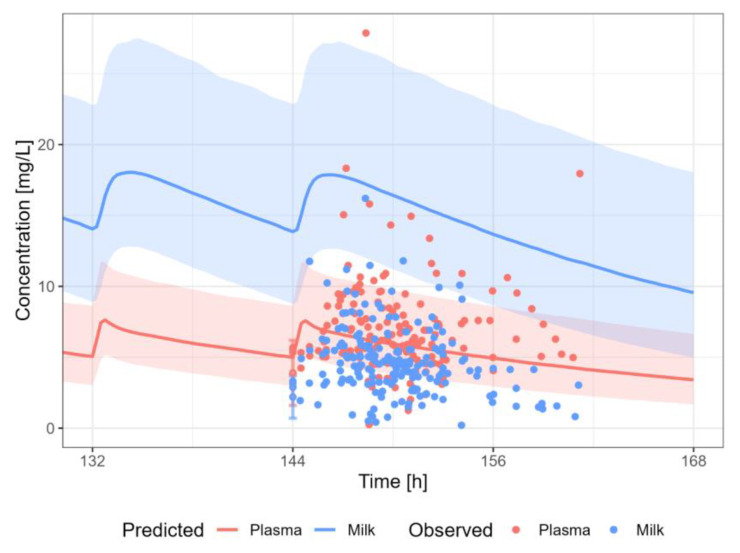
Median (solid line) and 5th–95th % prediction interval (shaded area) for oral administration of nevirapine 200 mg bidaily in plasma (red) and human milk (blue). Circles represent observed data from the literature [35,36,37,38,40,41,59,64]. Datapoints for which the time of sampling with respect to the last dose was not reported are indicated with open circles.

**Figure 11 pharmaceutics-15-01469-f011:**
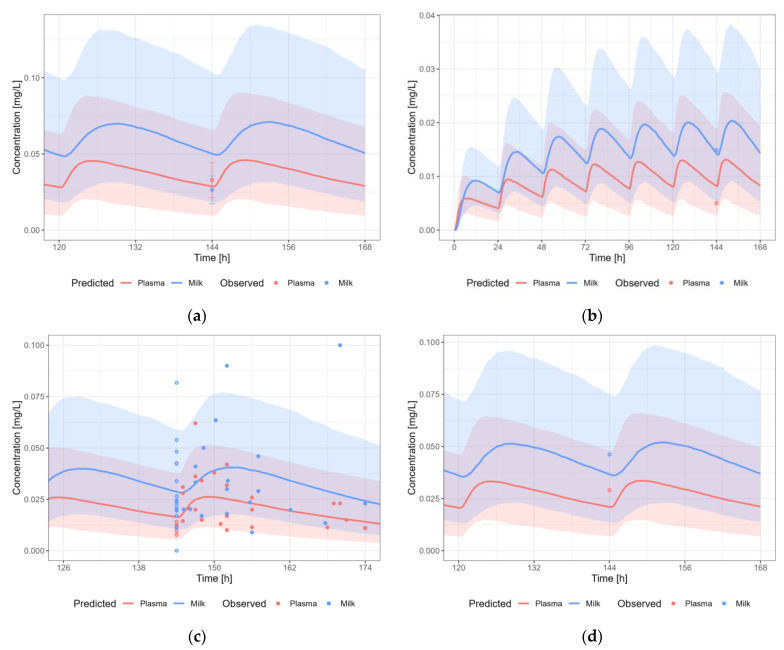
Median (solid line) and 5th–95th % prediction interval (shaded area) for: (**a**) oral administration of sertraline 87.5 mg/day [44]; (**b**) oral administration of sertraline 25 mg/day [46]; (**c**) oral administration of sertraline 50 mg/day [42,43,46,47,48,49,65,66]; and (**d**) oral administration of sertraline 64 mg/day [45] in plasma (red) and human milk (blue). Circles represent observed data from the literature with standard deviation (error bars) [42,43,44,46,47,48,49,65,66]. Datapoints for which the time of sampling with respect to the last dose was not reported are indicated with open circles.

**Figure 12 pharmaceutics-15-01469-f012:**
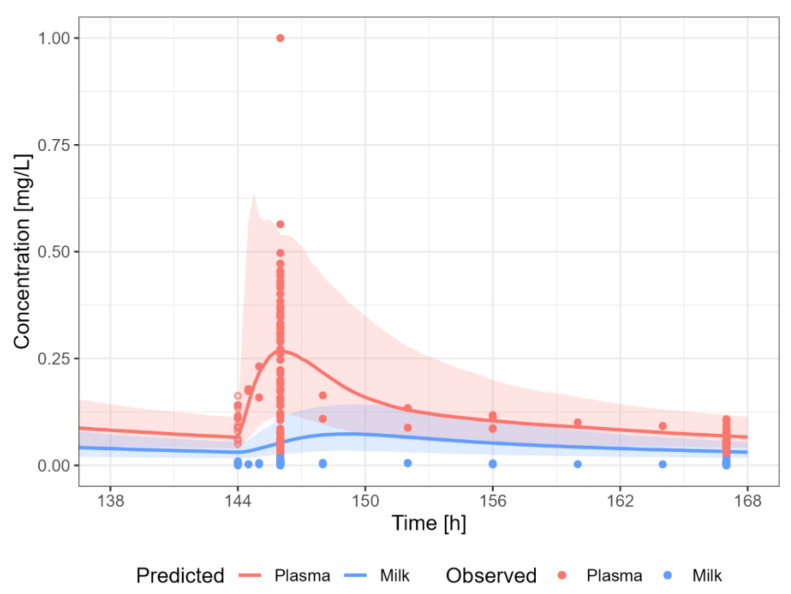
Median (solid line) and 5th–95th % prediction interval (shaded area) for oral administration of tenofovir 300 mg daily in plasma (red) and human milk (blue). Circles represent observed data from the literature [39,51,52,53,67]. Datapoints for which the time of sampling with respect to the last dose was not reported are indicated with open circles.

**Figure 13 pharmaceutics-15-01469-f013:**
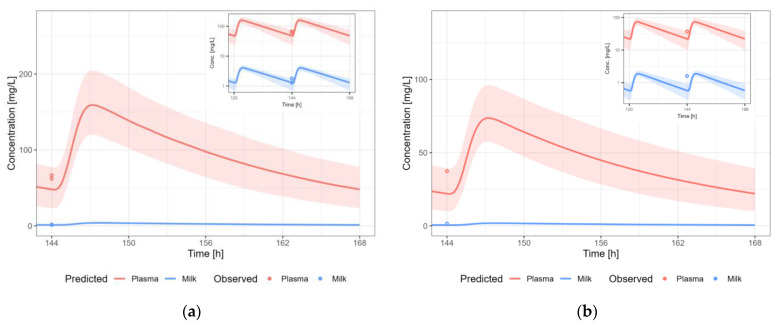
Median (solid line) and 5th–95th % prediction interval (shaded area) for: (**a**) oral administration of valproic acid 1500 mg/day [57]; (**b**) oral administration of valproic acid 11.3 mg/kg/day [54]; (**c**) oral administration of valproic acid 9.6 mg/kg/day [55]; and (**d**) oral administration of valproic acid 2100 mg/day [57] in plasma (red) and human milk (blue). Circles represent observed data from the literature [54,55,57]. Datapoints for which the time of sampling with respect to the last dose was not reported are indicated with open circles.

**Figure 14 pharmaceutics-15-01469-f014:**
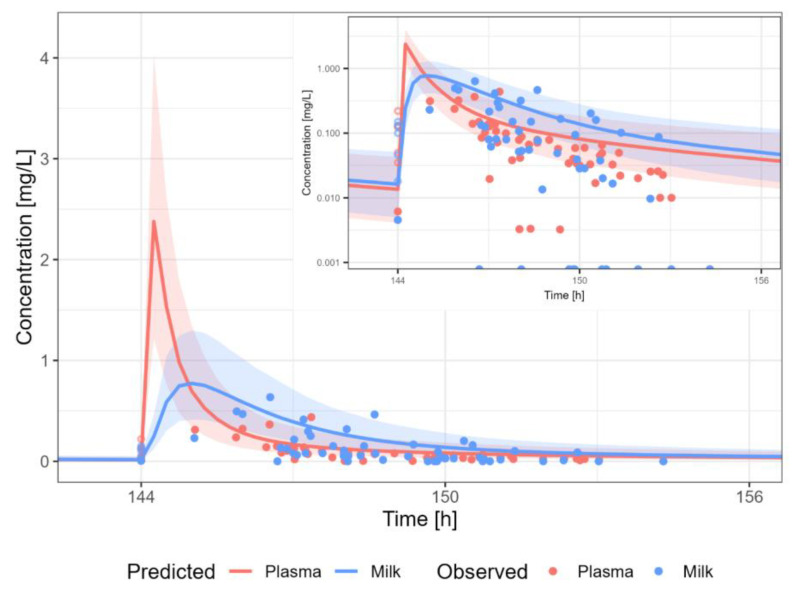
Median (solid line) and 5th–95th % prediction interval (shaded area) for oral administration of zidovudine 300 mg bidaily in plasma (red) and human milk (blue). Circles represent observed data from the literature [35,36,40,58,59]. Datapoints for which the time of sampling with respect to the last dose was not reported are indicated with open circles.

**Table 1 pharmaceutics-15-01469-t001:** Physicochemical properties of the ten model medicines.

Medicine	MW	BCS Class	pKa	LogP	HBD	HBA	PSA	f_u_	Main Elimination Route
Amoxicillin	365.40	I	3.23 (acid) 7.43 (base)	0.87	4	7	158	0.85	Renal
Caffeine	194.20	I	0.80 (base)	−0.07	0	3	58.44	0.70	Hepatic (CYP1A2)
Cetirizine	388.90	III	2.9 (acid) 8.0 (base) 2.2 (base)	1.50	1	5	53.00	0.07	Renal
Levetiracetam	170.21	I	- (neutral)	−0.60	1	2	63.40	0.90	Esterases
Metformin	129.16	III	2.80 (base) 11.50 (acid)	−1.43	3	1	91.50	1.00	Renal
Nevirapine	266.30	II	2.8 (base)	1.93	1	4	58.10	0.40	Hepatic (CYP3A4)
Sertraline	306.00	II	9.43 (base)	5.5	1	1	12.03	0.023	Hepatic (CYPs)
Tenofovir	287.21	III	1.35 (acid) 6.70 (acid) 3.80 (base)	1.87	3	8	136.38	0.993	Renal
Valproic acid	144.21	I	4.80 (acid)	2.75	1	2	37.30	0.14	Hepatic (UGTs)
Zidovudine	267.24	I	9.7 (acid)	0.05	2	6	108.30	0.80	Hepatic (UGT2B7)

Molecular weight (MW, g/mol); biopharmaceutical classification system (BCS); acid dissociation constant (pK_a_); lipophilicity as Log_10_ of the partition coefficient between octanol and water (LogP); hydrogen bound donors (HBD); hydrogen bound acceptors (HBA); polar surface area (PSA, Å^2^); fraction unbound in human plasma (f_u_). References to sources can be found in Appendix A.

**Table 2 pharmaceutics-15-01469-t002:** Predicted bidirectional clearance values between plasma and human milk.

Medicine	Secretion Clearance (mL/h)	Reuptake Clearance (mL/h)
Amoxicillin	46.90	263.47
Caffeine	824.02	603.21
Cetirizine	3031.27	1922.74
Levetiracetam	445.38	357.77
Metformin	21.73	138.37
Nevirapine	2413.86	1015.01
Sertraline	8925.70	3597.49
Tenofovir	51.96	129.00
Valproic acid	248.39	1423.15
Zidovudine	431.04	345.10

**Table 3 pharmaceutics-15-01469-t003:** Milk-to-plasma (M/P) ratio.

Medicine	Predicted M/P Ratio ^1^	Observed M/P Ratio	Reference
Amoxicillin	0.15	0.04–0.06 ^2^	[21]
Caffeine	0.95	0.52–1.16	[22,23,24]
Cetirizine	0.12	0.2 ^3^	[25,26]
Levetiracetam	1.11	0.46–1.79	[27,28,29,30]
Metformin	0.16	0.13–1.00	[31,32,33,34]
Nevirapine	2.68	0.2–1.5	[35,36,37,38,39,40,41]
Sertraline	1.62	0.12 ^4^–5.2	[42,43,44,45,46,47,48,49,50]
Tenofovir	0.40	0.025 ^4^–0.11	[39,51,52,53]
Valproic acid	0.03	0.013 ^4^–0.25	[54,55,56,57]
Zidovudine	1.10	0.3–3.21	[35,36,40,58,59]

^1^ M/P ratios were calculated based on predicted area-under-the-curve (AUC) in human milk and plasma. ^2^ For amoxicillin, only three single time point based M/P values were reported at the time of the peak concentration in plasma. However, the peak in human milk is delayed compared to plasma, potentially leading to an underestimation of the M/P ratio. Therefore, the M/P ratio was calculated using the peak concentration in plasma and the highest measured concentration in human milk. Alternatively, non-compartmental analysis was applied to estimate the area-under-the-curve (AUC) based M/P ratio, assuming that the elimination slope in human milk is identical to plasma. ^3^ Plasma concentrations in lactating women are not available. The M/P ratio was calculated using the observed steady-state AUC in human milk (0.50 mg*h/L), and the observed plasma AUC in non-lactating adults receiving the same dosing regimen (2.50 mg*h/L). ^4^ Some studies report human milk concentrations below the limit of quantitation.

**Table 4 pharmaceutics-15-01469-t004:** Predicted daily infant dosage (DID, mg/kg/day), and relative infant dose (RID, %).

Medicine	Daily Infant Dosage Based on Average Concentration (mg/kg/day) (Relative Infant Dose) (%)	Daily Infant Dosage Based on Maximal Concentration (mg/kg/day) (Relative Infant Dose) (%)
Amoxicillin	0.12 (0.24%)	0.19 (0.39%)
Caffeine	0.30 (5.98%)	0.41 (8.17%)
Cetirizine	0.002 (1.24%)	0.01 (3.62%)
Levetiracetam	6.16 (12%)	7.92 (16%)
Metformin	0.02 (0.10%)	0.02 (0.14%)
Nevirapine *	2.43 (37%)	2.78 (42%)
Sertraline	0.005 (0.63%)	0.01 (0.72%)
Tenofovir *	0.01 (0.15%)	0.01 (0.21%)
Valproic acid	0.52 (1.50%)	0.85 (2.44%)
Zidovudine	0.04 (0.40%)	0.12 (1.16%)

* DID (RID) should be interpreted with caution as human milk concentration-time profile was overpredicted.

**Table 5 pharmaceutics-15-01469-t005:** Daily infant dosage as percentage of common therapeutic infant dosages.

Medicine	Therapeutic Dosage Used as Reference (mg/kg/day)	Daily Infant Dosage Based on Average Concentration as Percentage of Therapeutic Dosage (%)	Daily Infant Dosage Based on Maximal Concentration as Percentage of Therapeutic Dosage (%)
Amoxicillin	50	0.24	0.39
Caffeine ^1^	5	5.96	8.13
Cetirizine	0.5	0.41	1.20
Levetiracetam	40	15.40	19.79
Metformin	-	-	-
Nevirapine ^2^	12	20.23	23.16
Sertraline	-	-	-
Tenofovir ^2^	6.5	0.12	0.16
Valproic acid	40	1.31	2.12
Zidovudine	24	0.17	0.48

^1^ Caffeine is administered only to preterm infants.; ^2^ Daily infant dosage and percentage of therapeutic dosage should be interpreted with caution as human milk concentration-time profile was overpredicted.

## Data Availability

The data presented in this study are openly available on Github (https://github.com/translatPK-KUL/LactationPBPK).

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
