# Peer review of "Generic Workflow to Predict Medicine Concentrations in Human Milk Using Physiologically-Based Pharmacokinetic (PBPK) Modelling—A Contribution from the ConcePTION Project"

_pharmaceutics, 2023, doi:10.3390/pharmaceutics15051469_

Round 1
Reviewer 1 Report
The academic and industrial ConcePTION consortium, mainly from Benelux, made a very good job in writing this important contribution. From a biostats/clinical epidemiology point of view, it has very well planned and executed. PKSim can be considered a good standard for these simulation trials, and the availability of a GitHub site for the readers does make sense, helping in a full comprehension of the take-home message. All the models have been estimated both for lactating and not-lactating adults, as well for infants. Probably, the criteria that drove the consortium to select these 10 molecules could be more deeply described. As for continuous covariates, being them much more frequent than categorical ones, I do recommend to always report them as median(IQR), all along the manuscript.
All the supplements, so well presented and dedicated to 9 different active pharmaceutical compounds and the last one to caffeine, could deserve futures single manuscripts! With compliments!
Author Response
Dear Reviewer 1,
We thank you for your time evaluating our manuscript. Below we would like to provide you with our answers to the suggestions made.
Point 1: The academic and industrial ConcePTION consortium, mainly from Benelux, made a very good job in writing this important contribution. From a biostats/clinical epidemiology point of view, it has very well planned and executed. PKSim can be considered a good standard for these simulation trials, and the availability of a GitHub site for the readers does make sense, helping in a full comprehension of the take-home message. All the models have been estimated both for lactating and not-lactating adults, as well for infants. Probably, the criteria that drove the consortium to select these 10 molecules could be more deeply described.
Response 1: The criteria for the selection of the model medicines were physicochemical diversity and the availability of clinical data in the lactating population. The physicochemical diversity was based on the chemical structures and the physicochemical properties, which are also described in Table 1. In addition, the BCS class of the medicines was added to this table (line 278). Furthermore, Tanimoto coefficients were calculated to show the (dis)similarity between the model medicines. The other criterion was the availability of clinical data in the lactating population, as this is required for evaluation of the predictive performance of the PBPK models. Clinical data collected from literature, while additional clinical data will become available from partners within the IMI ConcePTION project (i.e. clinical trials are ongoing for amoxicillin, cetirizine and metformin). A more detailed description was also added on line 107-115.
Point 2: As for continuous covariates, being them much more frequent than categorical ones, I do recommend to always report them as median(IQR), all along the manuscript.
Response 2: The interquartile range (IQR) was included for the continuous covariates. The median (IQR) breast volume was 1.0 (0.7-1.5) L (line 176 & 233). The specific blood flow rate in the breasts was 27.0 (26.7-27.3) mL/min/100g organ (line 176 & 233).
Point 3: All the supplements, so well presented and dedicated to 9 different active pharmaceutical compounds and the last one to caffeine, could deserve futures single manuscripts! With compliments!
Response 3: We thank the reviewer for this remark. We would also like to clarify that caffeine is also used as active pharmaceutical compound, for example in the treatment of apnea in premature infants.
Reviewer 2 Report
The manuscript describes the development of PBPK models including lactation physiology to predict drug concentrations in human milk and evaluation of model predictive performance using in vivo literature data. This issue is very important and timely, as little is known about milk levels of newer drugs of lactating women. The article is well written, however, several issues need clarification:
1. The criteria for drug selection should be clearly stated.
2. Some information (e.g. demographic data) about non-lactating adult individuals should be provided.
3. If male healthy adult and/or patient individuals were involved (l. 284), possible sex differences in PK of the studied drugs should be discussed.
4. It should be explained in the main text how the fractions in Figure 2 capture were estimated/obtained.
5. The main limitations of the current study should be indicated, e.g. a lack of prospective data to validate the models.
Author Response
Dear Reviewer 2,
We thank you for your time evaluating our manuscript. Below we would like to provide you with our answers to the questions and comments made.
The manuscript describes the development of PBPK models including lactation physiology to predict drug concentrations in human milk and evaluation of model predictive performance using in vivo literature data. This issue is very important and timely, as little is known about milk levels of newer drugs of lactating women. The article is well written, however, several issues need clarification:
Point 1: The criteria for drug selection should be clearly stated.
Response 1: The criteria for the selection of the model medicines were physicochemical diversity and the availability of clinical data in the lactating population. The physicochemical diversity was based on the chemical structures and the physicochemical properties, which are also described in Table 1. In addition, the BCS class of the medicines was added to this table (line 278). Furthermore, Tanimoto coefficients were calculated to show the (dis)similarity between the model medicines. The other criterion was the availability of clinical data in the lactating population, as this is required for evaluation of the predictive performance of the PBPK models. Clinical data were collected from literature, while additional clinical data will become available from partners within the IMI ConcePTION project (i.e. clinical trials are ongoing for amoxicillin, cetirizine and metformin). A more detailed description was also added on line 107-115.
Point 2: Some information (e.g. demographic data) about non-lactating adult individuals should be provided.
Response 2: We fully agree with this comment. A table with the demographic information, including the female ratio (see Point 3) was added in the supplementary files for each medicine.
Point 3: If male healthy adult and/or patient individuals were involved (l. 284), possible sex differences in PK of the studied drugs should be discussed.
Response 3: The PBPK model for “non-lactating” adults was developed using the the clinical data available in literature. Indeed, both female and male individuals were involved. The female ratio for each study was also added in the table with demographic information in the supplementary files. Unfortunately, we are limited by the (un)availablility of clinical data in literature. As only a signle study was found that studied the differences in PK between female and male patients for three medicines (nevirapine, sertraline and valproic acid) and no studies for the other medicines, this was not sufficient to thorougly evaluate whether the PBPK models were able to predict sex differences in PK. This point was also added as a limitation in the discussion of the manuscript on line 573-578: “First, the quality of the lactation PBPK models is dependent on the quality of the PBPK models for “non-lactating” adult individuals. The PBPK models were developed and evaluated using clinical data for both female and male individuals. Unfortunately, only a single study was found that studied the differences between female and male patients for three medicines (nevirapine, sertraline and valproic acid), and no studies were found for the other medicines. Therefore, it was not possible to fully evaluate the ability of the PBPK models to capture sex differences in PK. The PBPK models can be improved further when additional mechanistic insights (e.g. in vitro data for metabolism) or high-quality clinical data becomes available.“
Point 4: It should be explained in the main text how the fractions in Figure 2 capture were estimated/obtained.
Response 4: The workflow followed to add a breast compartment was based on the tutorial on how to extend a model to a special population in pregnancy by Dallmann et al. (2018). In the tutorial, it is explained how Dallmann et al. (2018) added a breasts compartment based on the heart compartment available in PK-Sim. We followed the same approach. Besides the reference to the tutorial, we also added a brief explanation in the main text on line 171-173: “Briefly, the breasts compartment (including physiological parameters, for example the fractions of the different sub-compartments) is a clone of the heart compartment available in PK-Sim..”
Point 5: The main limitations of the current study should be indicated, e.g. a lack of prospective data to validate the models.
Response 5: The lack of prospective data in the postpartum population is indeed a limitation, and was added in the discussion on line 553-555: “A limitation of the current study was the lack of prospective clinical data in the lactating population, to thoroughly evaluate the predictive performance of the PBPK models”. In addition, also the limitation related to sex differences in PK as discussed in point 3 was added in the discussion (line 573-578).
Reviewer 3 Report
This is a high quality paper. Almost no comments, just small suggestions:
1.Why you choose ten drugs as model ones, which selection criteria you applied. To which Biopharmaceutical Classification System (BCS) groups belong these drugs.
2. Breast volume (not all functional in milk production and drug transfer as a whole anatomical structure) or rather mammary gland should be rather used as the term
3. Why valproic acid and sertraline are out of 2-fold prediction error ratio.
4. Why you skip transporter information about, e.g. OCTs for metformin and OATs for tenofovir,
or nevirapine and valproic acid autoinduction, or caffeine CYP1A2 induction in smokers
5. line 222, 227 –Job et al. reference is not numbered.
Author Response
Dear Reviewer 3,
We thank you for your time evaluating our manuscript. Below we would like to provide you with our answers to the suggestions made.
This is a high quality paper. Almost no comments, just small suggestions:
Point 1: Why you choose ten drugs as model ones, which selection criteria you applied. To which Biopharmaceutical Classification System (BCS) groups belong these drugs.
Response 1: The criteria for the selection of the model medicines were physicochemical diversity and the availability of clinical data in the lactating population. The physicochemical diversity was based on the chemical structures and the physicochemical properties, which are also described in Table 1. In addition, the BCS class of the medicines was added to this table (line 278). Furthermore, Tanimoto coefficients were calculated to show the (dis)similarity between the model medicines. The other criterion was the availability of clinical data in the lactating population, as this is required for evaluation of the predictive performance of the PBPK models. Clinical data collected from literature, while additional clinical data will become available from partners within the IMI ConcePTION project (i.e. clinical trials are ongoing for amoxicillin, cetirizine and metformin). A more detailed description was also added on line 107-115.
Point 2: Breast volume (not all functional in milk production and drug transfer as a whole anatomical structure) or rather mammary gland should be rather used as the term
Response 2: The term breasts volume was used as this was also the term used in the tutorial by Dallmann et al. (2018) and in the lactating population that was used from Job et al. (2022). Briefly the breasts compartment was added in Mobi according to the tutorial by Dallmann et al. (2018). Next, the human milk was added as a compartment connected to the plasma of the breasts. The geometric mean volume of the human milk was 500 mL, in accordance with the semi-mechanistic model from Koshimichi et al. (2011) as this model was used to parametrize the transfer between plasma and human milk. Next, the structure was imported in PK -Sim for simulations. Here, it was combined with the population by Job et al. (2022), which includes physiological parameters (for example breasts volume) for the lactating population.
Point 3: Why valproic acid and sertraline are out of 2-fold prediction error ratio.
Response 3: The PBPK model for VPA was able to capture the pharmacokinetic behavior of the medicines in healthy volunteers and/or patients in 97 % of the simulations. For Nitsche and Mascher (1982), the plasma-concentration of 1 subject was not predicted well when 900 mg was given as 2 capsules. Importantly, the prediction for the same subject after 900 mg as three capsules or after administration of 1000 mg were good. In addition, the plasma concentration of the 5 other subjects were predicted within 2-fold for the three dosing regimens. The observed data also showed that the through concentrations of the last 2 doses before administration of the last dose decline, which might be explained by patient-specific autoinduction processes or a missed dose. Another study with different doses of intravenous administration was not predicted within 2-fold prediction error on AUC for one dose level. This study did not find an increase in plasma concentration with an increase in dose. Moreover, the AUC even declines when the dose is increased from 90 to 120 mg/kg and from 140 to 150 mg/kg. It is not clear what is the cause for these results, potentially complex non-linear kinetics are involved for valproic acid. The PBPK model for sertraline was able to adequately predict the AUC and Cmax for 89 % of the simulations. It is described in literature that there is a high inter-individual variability in the pharmacokinetic profiles, which is partly explained by the involvement of several enzymes (CYP2D6, CYP2B6, CYP3A4, CYP2C9, CYP2C19, and CYP2E1) with different genotypes in the metabolism. Further mechanistic insight (e.g. involvement of different genotypes for CYP enzymes) in the source of the variability could furhter improve the reference PBPK models. This was also described in the supplementary reports for sertraline and valproic acid, and briefly mentioned in the manuscript (line 310-317).
Point 4: Why you skip transporter information about, e.g. OCTs for metformin and OATs for tenofovir, or nevirapine and valproic acid autoinduction, or caffeine CYP1A2 induction in smokers
Response 4: Indeed, transporters can be considered to make the PBPK models more mechanistic. The autoinduction process for nevirapine was included in the PBPK model for CYP3A4, CYP1B6 and CYP2D6. For valproic acid, the autoinduction was considered but did not improve the predictions. The reason therefore is potentially the complex mechanism behind the autoinduction that is not fully understood yet, as not all clinical trials observe clear autoinduction. For caffeine, the template PBPK model available in PK-Sim was used. For metformin, a PBPK model from literature by Hanke et al. (2020) was used. Here OCT1/2, MATE1 and PMAT transport was included in the PBPK model. For tenofovir, we included tubular secretion in the kidney instead of OAT transport.
Overall, we decided to focus on the workflow and application for lactation. Relatively simple PBPK models were sufficient since they were able to describe the plasma-concentration time profiles adequately for the “non-lactating” population.
Point 5: line 222, 227 –Job et al. reference is not numbered.
Response 5: The reference for Job et al. is reference number [10], which is present on line 229.